# Mortality, morbidity and clinical care in a referral neonatal intensive care unit in Haiti

**Josie Valcin[1], Skenda Jean-Charles[1], Ana Malfa[2], Richard Tucker[3], Lindsay Dorcélus[4], Jacqueline Gautier[4], Michael P. Koster[1,5], Beatrice E. Lechner[1,3]***

**1** Warren Alpert Medical School of Brown University, Providence, Rhode Island, United States of America, **2** Brown University, Providence, Rhode Island, United States of America, **3** Department of Neonatology, Women & Infants Hospital, Providence, Rhode Island, United States of America, **4** Hôpital Saint-Damien/Nos Petits-Frères et Sœurs; Tabarre, Haiti, **5** Department of Pediatrics, Hasbro Children's Hospital, Providence, Rhode Island, United States of America

* blechner@wihri.org

**Data Availability Statement:** All relevant data are within the manuscript.

**Funding:** The authors received no specific funding for this work.

## Abstract

### Background

Neonatal mortality rates in Haiti are among the highest in the Western hemisphere. Few mothers deliver with a skilled birth attendant present, and there is a significant lack of pediatricians. The neonatal intensive care unit (NICU) at St. Damien Pediatric Hospital, a national referral center, is one of only five neonatology departments in Haiti. In order to target limited resources toward improving outcomes, this study seeks to describe clinical care in the St. Damien NICU.

### Methods

A retrospective medical record review was performed on available medical records on all admissions to the NICU between April 2016 and April 2017.

### Results

220 neonates were admitted to the NICU within the study epoch. The mortality rate was 14.5%. Death was associated with a maternal diagnosis of hypertension (p = 0.03) and neonatal diagnoses of lower gestational age (p<0.0001), lower birth weight (p<0.0001), prematurity (p = 0.002), RDS p = 0.01), sepsis (p<0.0001) and kernicterus (p = 0.04). The most common diagnoses were sepsis, chorioamnionitis, respiratory distress syndrome, jaundice, prematurity and perinatal asphyxia.

### Conclusions

This study demonstrates that preterm birth, sepsis, RDS and kernicterus are key contributors to neonatal mortality in a Haitian national pediatric referral center NICU and as such are promising interventional targets for reducing the neonatal mortality rate in Haiti.

**Competing interests:** The authors have declared that no competing interests exist.

## Introduction

Neonatal and infant mortality rates in Haiti are among the highest in the Western hemisphere, with rates that are markedly disparate from those of its neighboring country on the island of Hispaniola, the Dominican Republic. The current neonatal mortality rate was 32 per 1000 live births, compared to the Dominican Republic's rate of 16 per 1000 live births [1]. Of all deaths of children under the age of 5 in Haiti, 34% died within the first 28 days of life, the neonatal period [2]. Global Health Observatory data shows that between 2011–2017, only 41.6% of mothers delivered with a skilled birth attendant present [3]. The large percentage of women who deliver their infants without a skilled healthcare worker present and the significant shortage of neonatal intensive care units contribute to the high neonatal mortality rates [4]. Major causes of neonatal death include prematurity (34%), birth asphyxiation and trauma (26%), and sepsis/other infectious conditions (18%) [5]. In one study of a neonatal ICU in Haiti, sepsis accounted for 54.8% of admissions to the NICU between 2013 and 2015 [6].

Haiti is classified by the World Bank as low-income and is currently the poorest country in the Western Hemisphere, with more than half of Haitians living below the national poverty line [7]. Of the population of over ten million, one third are under the age of 15 [8]. While there are 23.5 physicians per 100,000 [9], the situation is even more dire for children. There are about 2.5 pediatricians per 100,000 children under the age of 15. In contrast, there are currently 101.2 certified pediatricians (including specialists) per 100,000 children in the United States [10]. The natural disasters that have occurred in the region have placed an additional strain on already limited healthcare services. There is a need for progressive, cost-effective health care practices that can alleviate both individual and governmental cost burden.

St. Damien Pediatric Hospital is a private non-profit hospital run by NPH/NPFS (Nuestros Pequeños Hermanos/Nos Petit Frères et Sœurs) in Tabarre, Haiti, supported through fundraising and charitable donations. The hospital was created in 1989, responding to need in the area for a hospital that could treat debilitating illnesses taking children's lives. Its mission is to not only care for the sick, but to offset the injustices of poverty and unemployment that make healthcare inaccessible for many individuals in the country. The hospital provides quality and dignified healthcare to children of any social level in emergent condition, and encourages parent participation through ongoing dialogue, educational opportunities, and offering any material support available to families.

The neonatal intensive care unit (NICU) at St. Damien Pediatric Hospital is one of only five neonatology departments currently operating in Haiti. The NICU was established after the devastating 2010 earthquake, when damage to other hospitals in the area exacerbated the need for specialized neonatal care. The NICU currently has twenty beds and is staffed with pediatric physicians with neonatal training, as well as trained neonatal nurses, who provide specialized bedside care to the neonatal patients.

This study seeks to describe clinical care and outcomes in the St. Damien NICU, a national referral center NICU in Haiti, in order to target limited resources toward improving outcomes.

## Materials and methods

### Data collection

A quantitative, descriptive, retrospective medical record review was performed on available medical records within the study timeframe. Haitian Institutional Review Board approval was obtained through Comité National de Bioéthique, which waived the requirement for informed consent. Data were collected from the medical records of newborns admitted to the NICU at

St. Damien Pediatric Hospital between April 2016 and April 2017. Data were fully anonymized upon accessing and collection from the medical record. Inclusion criteria for the study included every infant admitted to the St. Damien NICU during the study period. The admissions book in the NICU was used to identify the names, birth date, birthweight, gestational age, mortality, and chart number of neonates. Typically, this information is recorded at time of admission or at death. Then, charts were physically identified and data transcribed in the hospital archives. Data collection included maternal and infant demographics, diagnoses, laboratory results, interventions and therapies. Due to flooding of basement archives, where charts of deceased patients were kept, some charts of neonates who had died were unable to be reviewed. These infants were excluded from the study, which in turn decreased the calculated mortality rate.

### Data analysis

Descriptive statistics for the sample as a whole and by survival are shown. Comparisons between deaths and survivors were made by t-tests for normally distributed continuous variables and Wilcoxon tests when the data were non-normally distributed. Categorical variables were analyzed using the chi-square test, or in the case of small cell sizes, Fisher's exact test. Statistical significance was defined as a two-sided p-value $< 0.05$.

## Results

A total of 220 neonates admitted to the St. Damien Pediatric Hospital NICU were identified within the study epoch.

Of these, 32 (14.5%) died during the hospitalization. Those who died were more likely to have a maternal diagnosis of hypertension compared to those who survived to discharge, as well as lower gestational age, lower birth weight, and shorter length of hospitalization. The diagnoses associated with an increased outcome of death included prematurity, RDS, sepsis and kernicterus. Diagnoses associated with a decreased outcome of death were chorioamnionitis, transient tachypnea of the newborn (TTN) and jaundice (Table 1).

Most neonates were inborn. Over half of births occurred via Cesarean section and the mean ten minute Apgar score was 7.8 (Table 1).

During the study period, a total of 20 (10%) parents and their neonates left the hospital against medical advice (Table 1).

The most common diagnosis was sepsis, at 91.5%. Other common diagnoses were chorioamnionitis, respiratory distress syndrome, jaundice, prematurity and perinatal asphyxia (Table 1). Although prematurity is defined as a gestational age of less than 37 weeks, the diagnosis of prematurity was also made using the Ballard score, given unreliable dating in the setting of low rates of prenatal care and access to dating ultrasound.

Endotracheal intubation and conventional ventilator support were rare, while respiratory support, which 60% of admissions received, most commonly consisted of nasal CPAP and nasal cannula (Table 2). Intubation was associated with death (total intubations n = 8; intubation among infants who died n = 6; p = 0.003).

Most infants received breastmilk, with few receiving breastmilk plus formula, and only one receiving formula alone (Table 2).

55% of neonates received phototherapy, and 19% received a sun bath, in which infants are placed in the sun in a cot as a therapy for jaundice (Table 2). Mean total bilirubin was 8.33.

Almost all (99%) of admitted infants received ampicillin and 97.3% received gentamicin during the course of their hospitalization (Table 2).

**Table 1. Maternal and neonatal characteristics.**

| Characteristics (mean ± SD; n (%)) | All admissions n = 220 | Survivors n = 188 (85.0%) | Deaths n = 32 (14.5%) | Pᵃ |
|---|---|---|---|---|
| **Maternal** | | | | |
| Age (n = 185) | 29.5 ± 6.9 | 29.6 ± 6.9 | 28.7 ± 6.6 | 0.54 |
| Gravida (n = 187) | 2.3 ± 1.6 | 2.2 ± 1.5 | 2.8 ± 2.3 | 0.59 |
| Para (n = 186) | 1.4 ± 1.4 | 1.3 ± 1.2 | 1.9 ± 2.1 | 0.53 |
| Cesarean section (n = 220) | 115 (52.3) | 100 (57.1) | 14 (46.7) | 0.50 |
| HIV (n = 220) | 3 (1.4) | 1 (0.54) | 2 (6.3) | 0.058 |
| Syphilis (n = 220) | 3 (1.4) | 2 (1.1) | 1 (3.2) | 0.37 |
| Alcohol use (n = 220) | 2 (0.9) | 2 (1.1) | 0 (0) | 1.00 |
| Tobacco use (n = 220) | 0 (0) | 0 (0) | 0 (0) | n/a |
| Drug use (n = 220) | 1 (0.5) | 1 (0.54) | 0 (0) | 1.00 |
| Hypertension (n = 220) | 25 (12.9) | 17 (10.4) | 8 (27.6) | 0.03 |
| **Infant** | | | | |
| Gestational age (n = 218) | 36.4 ± 4.1 | 37.1 ± 3.5 | 32.1 ± 4.8 | <0.0001 |
| Female (n = 220) | 103 (46.8) | 84 (44.9) | 18 (56.3) | 0.25 |
| Birth weight (n = 219) | 2304 ± 888 | 2427 ± 830 | 1597 ± 906 | <0.0001 |
| Singleton (n = 220) | 206 (93.6) | 176 (94.1) | 29 (90.6) | 0.44 |
| Inborn (n = 220) | 216 (98.8) | 183 (97.9) | 32 (100) | 1.00 |
| Apgar 1 min (n = 186) | 6.1 ± 1.8 | 6.1 ± 1.8 | 5.7 ± 2.0 | 0.35 |
| Apgar 5 min (n = 187) | 7.2 ± 1.6 | 7.2 ± 1.6 | 6.8 ± 1.6 | 0.19 |
| Apgar 10 min (n = 170) | 7.8 ± 2.0 | 7.8 ± 2.0 | 7.3 ± 2.1 | 0.09 |
| Length of hospitalization (n = 220) | 14 ± 14.4 | 15.3 ± 15.0 | 6.5 ± 7.1 | <0.0001 |
| **Neonatal diagnoses** | | | | |
| Preterm | 79 (35.9) | 56 (30.0) | 19 (59.3) | 0.002 |
| Chorioamnionitis | 156 (70.9) | 141 (75.4) | 15 (46.9) | 0.003 |
| Sepsis | 199 (91.5) | 18 (9.6) | 17 (53.1) | <0.0001 |
| Respiratory distress syndrome (RDS) | 91 (41.4) | 71 (38.0) | 20 (62.5) | 0.01 |
| Transient tachypnea of the newborn | 21 (9.5) | 20 (10.7) | 0 (0) | 0.05 |
| Pneumonia | 16 (7.4) | 6 (3.2) | 1 (3.1) | 1.00 |
| Meconium aspiration syndrome | 19 (13.4) | 12 (6.4) | 2 (6.3) | 1.00 |
| Bronchopulmonary dysplasia | 3 (1.4) | 0 (0) | 1 (3.1) | 0.15 |
| Perinatal asphyxia | 62 (28.2) | 55 (29.4) | 10 (31.2) | 0.84 |
| Encephalopathy | 11 (5.1) | 0 (0) | 1 (3.1) | 0.15 |
| Seizures | 16 (7.4) | 3 (1.6) | 0 (0) | 1.00 |
| Meningitis | 5 (2.3) | 1 (0.5) | 0 (0) | 1.00 |
| Hydrocephalus | 2 (0.9) | 1 (0.5) | 1 (3.1) | 0.27 |
| Intraventricular hemorrhage (IVH) | 4 (1.1) | 2 (1.1) | 2 (6.3) | 0.10 |
| Kernicterus | 6 (1.6) | 3 (1.6) | 3 (9.4) | 0.04 |
| Jaundice | 81 (36.8) | 116 (62.0) | 11 (34.4) | 0.006 |
| Congenital heart disease | 10 (4.7) | 14 (7.5) | 2 (6.3) | 1.00 |
| Kidney disease | 3 (1.6) | 3 (1.6) | 0 (0) | 1.00 |
| Necrotizing enterocolitis | 10 (4.7) | 5 (2.7) | 1 (3.1) | 1.00 |
| Anemia | 34 (16.9) | 31 (16.6) | 3 (9.4) | 0.43 |
| Patients who left AMA (n = 200) | 20 | 20 | n/a | |

Abbreviations: AMA = against medical advice

ᵃSurvivors vs. Deaths

**Table 2. Interventions.**

| Respiratory | N = 220 (%) |
|---|---|
| Intubation | 8 (4) |
| Ventilation | 129 (60) |
| CPAP (n = 130) | 68 (52) |
| Nasal cannula (n = 130) | 61 (47) |
| Conventional ventilation (n = 130) | 1 (1) |
| **Nutrition** | **N = 181 (%)** |
| Breast milk | 152 (84) |
| Formula | 1 (0.6) |
| Breast milk and formula | 28 (16) |
| **Hyperbilirubinemia** | **N = 220 (%)** |
| Phototherapy | 118 (54) |
| Sunbath | 40 (18) |
| | **Mean ± SD** |
| Total bilirubin (n = 114) | 8.33 ± 4.3 |
| Direct bilirubin (n = 114) | 1.65 ± 1.53 |
| **Medication** | **N = 220 (%)** |
| Ampicillin | 218 (99) |
| Gentamicin | 214 (97.3) |
| Ranitidine | 173 (78.6) |
| Cefotaxime | 121 (55.0) |
| Phenobarbital | 114 (51.8) |
| Caffeine Citrate | 78 (35.5) |
| Sodium bicarbonate | 62 (28.2) |
| Ceftazidime | 44 (20.0) |
| Metronidazole | 41 (18.6) |
| Phenytoin | 24 (10.9) |

Abbreviations: CPAP = continuous positive airway pressure

## Discussion

Neonatal mortality and morbidity are increased in low resource settings, including Haiti [11]. In order to target limited resources toward improving outcomes, we performed a chart review of a national referral NICU in Haiti to ascertain areas for targeted interventions. Similar chart reviews in other low and mid income countries (LMICs), such as Uganda [12] have been performed and have shown that evaluation of current practices may lead to decreased mortality in low resource settings such as in Rwanda [13]. The differences in characteristics observed between the neonates who died in the NICU and those that survived to discharge suggest specific patterns of neonatal morbidity and mortality.

Maternal hypertension may lead to a decision to deliver an infant, subsequently leading to prematurity and thus lower gestational age and lower birth weight. Prematurity, in turn, is associated with an increased risk of RDS [14]. Meanwhile, sepsis is associated with prematurity in general, but is also well documented as a common cause of neonatal mortality in low resource settings [15]. Similarly, kernicterus is a complication of prematurity that is virtually nonexistent in otherwise healthy neonates in high resource settings but is prevalent in LMICs [16]. A shorter length of hospitalization is likely a reflection of neonates who die early during the course of hospitalization. The diagnoses associated with a decreased outcome of death, on

the other hand, chorioamnionitis, TTN and jaundice, may be a reflection of these characteristics indicating a less severe manifestation of the same pathophysiology that the diagnoses associated with higher death rates, sepsis, RDS and kernicterus, represent. Additional factors that may affect length of hospitalization, in addition to diagnosis, may include socioeconomic factors, such as parental ability to take the infant home and care for them outside the hospital. The high number of deaths associated with intubation and the low number of intubated infants is reflective of the lack of access to an adequate number of ventilators and related supplies.

St. Damien Pediatric Hospital is a national pediatric referral center that also houses a labor and delivery suite. Because of previous nosocomial infectious outbreaks, the NICU is restricted to neonates who were born at St. Damien. A separate NICU for children born outside of St. Damien is currently being developed. Thus, most patients are inborn. The high Cesarean section rate is similar to rates in other limited resource settings, such as the Dominican Republic [17]. Given the fact that less than half of births occur in the presence of a trained birth attendant [3], women are more likely to present to the hospital without adequate prenatal care or during complicated labor after attempting home birth, thus leading to referral centers such as St. Damien Pediatric Hospital having a higher risk population. Although high perinatal asphyxia rates may be secondary to delivery practices, they are more likely also a reflection of this high risk patient population. Increasing the number of births that are attended by trained birth attendants in the community may be a viable strategy to decrease the rates of both Cesarean sections and perinatal asphyxia.

It is interesting to note the 10% rate of parents leaving the NICU with their infant against medical advice. A Saudi Arabian study demonstrated a rate of 1.6% [18], while a study in India showed a rate of 25.4% [19]. In the latter study, economic considerations were the most commonly indicated reason by parents, followed by lack of improvement and poor prognosis [19]. The most plausible explanation for this phenomenon within the Haitian economic and cultural framework is a lack of resources and the complex socioeconomic situation of families in addition to the medical challenges their infants face.

Sepsis was the most common diagnosis noted, and antibiotic usage was high. While multiple factors leading to increased rates of sepsis and mortality from sepsis have been documented in LMICs [15], evaluating these outcomes may provide direction in prioritizing effective countermeasures. The high rates of antibiotic use compared to rates in high resource settings such as 17% in a study of US NICUs [20] are likely a reflection of the sepsis rates.

Jaundice is another common diagnosis in the St. Damien NICU, with a rate of kernicterus that is higher than in high resource settings, such as a rate of 1.3 per 100,000 in a Swedish population based study [21]. Standard phototherapy was a common intervention, as well as sun baths when phototherapy was not available. Given the dangers associated with sun bath usage, the procurement of adequate phototherapy lights would likely lead to more appropriate therapeutic interventions.

An interesting observation is the very high rate of breastfeeding and exclusive breastfeeding compared to high resource settings. In the US, 83.8% of newborns were ever breastfed, 47.5% were exclusively breastfed at 3 months of age [22], and breastfeeding initiation in a NICU was 85% [23]. This may be secondary to the lack of access to newborn formula and thus a higher dependence on mother's milk in the Haitian population.

A limitation of the study is that it is a retrospective chart review, and thus the data quality is lower than it would be had the data been collected prospectively, given the inherent limitation associated with data extracted from existing medical records. Thus, the internal validity of the study is not 100%. Nonetheless, all medical records that were available were reviewed, so the data represent as accurate a description of NICU patients during the study timeframe as is

possible. The results of this study cannot be extrapolated to all of Haiti given its limitations, and thus external validity is also not 100%. However, because St. Damien Pediatric Hospital plays a unique role in Haitian healthcare for neonates as a national pediatric referral center, this study, which has elucidated outcomes for neonates beyond what was previously known about the state of neonatal care in Haiti, is a reflection of the highest level of neonatal care available in Haiti and as such is a useful tool in the development of effective neonatal interventions in Haiti.

As a next step, randomized studies of targeted neonatal interventions are necessary to achieve improved outcomes. In a trial in Guatemala, identifying women at risk for preterm birth, administering corticosteroids and encouraging delivering at a health care facility, reduced neonatal mortality [24].

Additional areas not reflected in this study include larger structures that can affect neonatal health care, such as sociopolitical climate, resource supply, and financial strain that are connected to the hospital's functioning and thus, neonatal health outcomes.

## Conclusions

In summary, this study demonstrates that preterm birth, sepsis, RDS and kernicterus are key contributors to neonatal mortality in a Haitian national pediatric referral center NICU and as such are promising interventional targets for reducing the neonatal mortality rate in Haiti.

## Author Contributions

**Conceptualization:** Josie Valcin, Skenda Jean-Charles, Lindsay Dorcélus, Jacqueline Gautier, Michael P. Koster, Beatrice E. Lechner.

**Data curation:** Josie Valcin, Skenda Jean-Charles, Ana Malfa, Lindsay Dorcélus.

**Formal analysis:** Josie Valcin, Skenda Jean-Charles, Ana Malfa, Richard Tucker, Michael P. Koster, Beatrice E. Lechner.

**Investigation:** Josie Valcin, Skenda Jean-Charles, Ana Malfa, Lindsay Dorcélus, Jacqueline Gautier, Beatrice E. Lechner.

**Methodology:** Josie Valcin, Skenda Jean-Charles, Ana Malfa, Richard Tucker, Lindsay Dorcélus, Jacqueline Gautier, Michael P. Koster, Beatrice E. Lechner.

**Project administration:** Jacqueline Gautier, Beatrice E. Lechner.

**Resources:** Beatrice E. Lechner.

**Supervision:** Jacqueline Gautier, Michael P. Koster, Beatrice E. Lechner.

**Writing – original draft:** Josie Valcin, Skenda Jean-Charles, Beatrice E. Lechner.

**Writing – review & editing:** Josie Valcin, Skenda Jean-Charles, Ana Malfa, Richard Tucker, Lindsay Dorcélus, Jacqueline Gautier, Michael P. Koster, Beatrice E. Lechner.

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
