## [Decision Letter · Decision Letter 0]

19 Aug 2020

PONE-D-20-12677

Mortality, morbidity and clinical care in a referral neonatal intensive care unit in Haiti

PLOS ONE

Dear Dr. Beatrice Lechner,

Thank you for submitting your manuscript to PLOS ONE. After careful consideration, we feel that it has merit but does not fully meet PLOS ONE’s publication criteria as it currently stands. Therefore, we invite you to submit a revised version of the manuscript that addresses the points raised during the review process.

We look forward to receiving your revised manuscript.

Kind regards,

Victor Adekanmbi, M.D., PhD

Academic Editor

PLOS ONE

2. In the ethics statement in the manuscript and in the online submission form, please provide additional information about the patient records used in your retrospective study. Specifically, please ensure that you have discussed whether all data were fully anonymized before you accessed them and/or whether the IRB or ethics committee waived the requirement for informed consent. If patients provided informed written consent to have data from their medical records used in research, please include this information.

Reviewers' comments:

Reviewer's Responses to Questions

**Comments to the Author**

1. Is the manuscript technically sound, and do the data support the conclusions?

Reviewer #1: Partly

Reviewer #2: Yes

2. Has the statistical analysis been performed appropriately and rigorously? 

Reviewer #1: Yes

Reviewer #2: Yes

3. Have the authors made all data underlying the findings in their manuscript fully available?

Reviewer #1: Yes

Reviewer #2: Yes

4. Is the manuscript presented in an intelligible fashion and written in standard English?

Reviewer #1: No

Reviewer #2: Yes

5. Review Comments to the Author

Reviewer #1: Manuscript Number: PONE-D-20-12677

A- Summary

Josie et al. aim at describing the clinical care and services provided at St Damien Neonatal Intensive Care Unit (NICU). To do so, they conducted a review of existing patient medical records over one year from April 2016 to Avril 2017. They used descriptive statistics including student and chi-square or Fisher exact tests to compare means and proportions, respectively. After describing the most common diagnosis by outcome status and the captured clinical interventions, they concluded that the findings provided an opportunity to develop targeting strategies to improve neonatal outcomes in Haiti.

Although there are significant gaps in the literature in describing the clinical features of NICU in Haiti, I have noticed critical methodological issues that should be addressed before this study can be considered as eligible for publication.

B- Major issues

• The method section only contains a few words on the statistical analyses performed. While they provided enough information on the study site in the background section, more information should be provided on the following elements:

a. The study design;

b. inclusion and exclusion criteria;

c. the outcome definition and calculation; need to be defined;

d. and data quality (Given the inherent limitation associated with data extracted from existing medical records);

• The way the results were being interpreted suggest an analytic study that would aim to establish the risk factors to neonatal death at the study site, which would require other types of study designs and statistical analysis. For instance, the following sentence clearly suggests that a relative risk has been estimated to establish the association between the risk factors and the outcomes: "The diagnoses associated with an increased risk of death included prematurity, RDS, sepsis, and kernicterus. Diagnoses associated with a decreased risk of death were chorioamnionitis, transient tachypnea of the newborn (TTN), and jaundice (Table 1)". Another example is this sentence: "Intubation was associated 158 with death (total intubations n=8; intubation among infants who died n=6; 159 p=0.003)". The most important point here is the need to keep the result interpretation consistent with the study objectives and methods.

• The study limitations that are subject to affect both internal and external validity should be clearly stated and particularly those inherent to the study design itself;

• The conclusion seems to suggest that these findings could be extrapolated to the whole country should be reconsidered once the study limitations are being clearly stated;

C- Minor issues

• The authors need to make sure that the most recent pieces of evidence are being considered. For instance, the current neonatal mortality rate in Haiti is 32 per 100 live birth (https://www.dhsprogram.com/pubs/pdf/FR326/FR326.pdf) instead of 25 per 100 live birth, as mentioned.

• Need an in-text citation for this sentence: "The large percentage of women who deliver their infants without a skilled healthcare worker present and the significant shortage of neonatal intensive care units contributes to the high neonatal mortality rates"; If available, it would be great if more socio-demographic information (maybe characteristics of the mothers) can be provided in table one beyond age;

• The percentage for the other diagnosis needs to be provided, including the n value "The most common diagnosis was sepsis, at 91.5%. Other common diagnoses 149 were chorioamnionitis, respiratory distress syndrome, jaundice, prematurity, and 150 perinatal asphyxia (Table 1)";

• It would be great if the author could present table 2 according to the neonates' outcomes as well;

• The authors should describe how data quality may have impacted the outcome (neonatal death) measurement (Potential information bias);

• The authors should explore other factors that can affect the Length of Stay in NICU;

Given both the major and minor issues listed above, I do not recommend publishing this article as submitted. Significant modification in the method and the conclusion sections will be needed to address the issues listed. It seems like the authors have enough variable to estimate a GLM to more highlight the roles of the contributors discussed in the neonatal death rate at the study site while clearly state the study limitations.

Reviewer #2: The paper is well articulated- the study purpose, methodology, data collection and analysis are well documented. The results of the study are also well documented and were performed to a high technical standard with sufficient detail. The results of this study have not been published elsewhere. The focus of this study represents findings that would be similar in most low resource setting countries. The authors have ably described the causes of neonatal mortality in a facility in Haiti and include causes such as maternal hypertension leading to low gestational age, low birth weight, prematurity, jaundice among others. Data was collected from medical records and was analyzed using T-test and wilcoxon tests and chi square. Overall 32 out of 220 neonates died representing 14.5%. The authors were able to provide conclusions in a comprehensive manner and the article is presented in a scholarly manner with high academic standard. The research therefore meets all required standards for publication. There is evidence of ethical approval and research integrity. However, despite mentioning the issue of low Skilled Birth Attendants and low number of pediatricians, the authors did not discuss the role of neonatal nurses in such a health care setting as they would normally spend more time in the clinical setting.

6. PLOS authors have the option to publish the peer review history of their article (what does this mean?). If published, this will include your full peer review and any attached files.

Reviewer #1: No

Reviewer #2: **Yes: **Professor Address Malata

---

## [Author Response · Author response to Decision Letter 0]

11 Sep 2020

We thank the reviewers for their encouraging comments and their thorough and helpful critique. We have carefully considered their comments and have made changes based on them that we believe improve the overall manuscript. Most noteworthy is the addition of further details in Methods as well as the addition of study limitations to the Discussion. Please find below a point-by-point response to the Reviewer’s critique.

Reviewer 1

Critique 1: “The method section only contains a few words on the statistical analyses performed. While they provided enough information on the study site in the background section, more information should be provided on the following elements: a. The study design; b. inclusion and exclusion criteria; c. the outcome definition and calculation; need to be defined; d. and data quality (Given the inherent limitation associated with data extracted from existing medical records).”

We have added additional information on the statistical analysis to the methods section (lines 130-135), as well as further clarifications on points a and b (lines 110, 116-117, 125-126). We added a paragraph on point d to the Discussion (lines 226-279). Given that this study is a descriptive study, an outcome definition or calculation is not appropriate.

Critique 2: “The way the results were being interpreted suggest an analytic study that would aim to establish the risk factors to neonatal death at the study site, which would require other types of study designs and statistical analysis. For instance, the following sentence clearly suggests that a relative risk has been estimated to establish the association between the risk factors and the outcomes: "The diagnoses associated with an increased risk of death included prematurity, RDS, sepsis, and kernicterus. Diagnoses associated with a decreased risk of death were chorioamnionitis, transient tachypnea of the newborn (TTN), and jaundice (Table 1)". Another example is this sentence: "Intubation was associated 158 with death (total intubations n=8; intubation among infants who died n=6; 159 p=0.003)". The most important point here is the need to keep the result interpretation consistent with the study objectives and methods.”

In order to clarify the objectives of the study and thus the interpretation of the results more clearly, we have edited the abstract and background (lines 32-34, 102-104) to state that this study seeks to describe clinical care in the St. Damien NICU in order to target limited resources toward improving outcomes. We did not calculate relative risks. In order to avoid confusion, we changed the word “risk” to “outcome” to more accurately reflect our analysis (lines 146-147, 209). Finally, this retrospective cohort study is adequate to describe the relationship of characteristics with death, without positing that these are causes.

Critique 3: “The study limitations that are subject to affect both internal and external validity should be clearly stated and particularly those inherent to the study design itself.”

We have added a discussion of internal and external validity to the Discussion (lines 266-279).

Critique 4: “The conclusion seems to suggest that these findings could be extrapolated to the whole country should be reconsidered once the study limitations are being clearly stated.”

We have clarified in the Discussion (lines 266-279) that there are limitations to the study that preclude extrapolating to all of Haiti based on this study.

Critique 5: “The authors need to make sure that the most recent pieces of evidence are being considered. For instance, the current neonatal mortality rate in Haiti is 32 per 100 live birth (https://www.dhsprogram.com/pubs/pdf/FR326/FR326.pdf) instead of 25 per 100 live birth, as mentioned.”

Thank you for bringing the more current citation to our attention. We have updated the manuscript (line 60).

Critique 6: “Need an in-text citation for this sentence: "The large percentage of women who deliver their infants without a skilled healthcare worker present and the significant shortage of neonatal intensive care units contributes to the high neonatal mortality rates"; If available, it would be great if more socio-demographic information (maybe characteristics of the mothers) can be provided in table one beyond age.”

We have added a citation for the sentence above (line 63). Unfortunately, additional demographic information regarding the mothers beyond what was presented in Table 1 was not available in the charts.

Critique 7: “The percentage for the other diagnosis needs to be provided, including the n value "The most common diagnosis was sepsis, at 91.5%. Other common diagnoses 149 were chorioamnionitis, respiratory distress syndrome, jaundice, prematurity, and perinatal asphyxia (Table 1).”

The percentages, as well as the n values, for each diagnosis, are listed in Table 1.

Critique 8: “It would be great if the author could present table 2 according to the neonates' outcomes as well.”

We weighed this suggestion in depth. While Table 1 was structured according to the outcomes of death vs. survival so as to demonstrate which pre-existing patient characteristics were associated with death, the role of Table 2 was to show which interventions were available in the NICU and utilized. Thus, demonstrating outcomes according to interventions utilized would not result in meaningful metrics.

Critique 9: “The authors should describe how data quality may have impacted the outcome (neonatal death) measurement (Potential information bias).”

We have included this information in the Methods section (line 125).

Critique 10: “The authors should explore other factors that can affect the Length of Stay in NICU.”

We have included an exploration of this topic under Discussion (lines 212-215).

Critique 11: “It seems like the authors have enough variable to estimate a GLM to more highlight the roles of the contributors discussed in the neonatal death rate at the study site while clearly state the study limitations.”

While there may be enough deaths in the data to do a GLM (generalized linear model/logistic regression), those models usually involve designating a particular variable as a focal cause of death and include other variables as possible confounders. This is a descriptive study looking at associations of a number of different outcomes relating to death, not one particular thing. Thus, a GLM would not add useful information.

Reviewer 2

Critique 1: “However, despite mentioning the issue of low Skilled Birth Attendants and low number of pediatricians, the authors did not discuss the role of neonatal nurses in such a health care setting as they would normally spend more time in the clinical setting.”

Thank you for pointing out this oversight on our part. In addition to birth attendants and pediatricians, neonatal nurses play a central role in providing care to neonates. We have added this information to the Introduction (lines 98-100).

---

## [Decision Letter · Decision Letter 1]

28 Sep 2020

Mortality, morbidity and clinical care in a referral neonatal intensive care unit in Haiti

PONE-D-20-12677R1

Dear Dr. Beatrice Lechner,

We’re pleased to inform you that your manuscript has been judged scientifically suitable for publication and will be formally accepted for publication once it meets all outstanding technical requirements.

Kind regards,

Victor Adekanmbi, M.D., PhD

Guest Editor

PLOS ONE

---

## [Editor Report · Acceptance letter]

5 Oct 2020

PONE-D-20-12677R1 

Mortality, morbidity and clinical care in a referral neonatal intensive care unit in Haiti 

Dear Dr. Lechner:

I'm pleased to inform you that your manuscript has been deemed suitable for publication in PLOS ONE. Congratulations! Your manuscript is now with our production department. 

Kind regards, 

on behalf of

Dr Victor Adekanmbi 

Guest Editor

PLOS ONE